# Genomic portrait and relatedness patterns of the Iron Age Log Coffin culture in northwestern Thailand

Selina Carlhoff [1] ✉, Wibhu Kutanan[2], Adam B. Rohrlach [1,3], Cosimo Posth [4,5], Mark Stoneking[6,7], Kathrin Nägele [1], Rasmi Shoocongdej [8,9] ✉ & Johannes Krause [1] ✉

The Iron Age of highland Pang Mapha, northwestern Thailand, is characterised by a mortuary practice known as Log Coffin culture. Dating between 2300 and 1000 years ago, large coffins carved from individual teak trees have been discovered in over 40 caves and rock shelters. While previous studies focussed on the cultural development of the Log Coffin-associated sites, the origins of the practice, connections with other wooden coffin-using groups in Southeast Asia, and social structure within the region remain understudied. Here, we present genome-wide data from 33 individuals from five Log Coffin culture sites to study genetic ancestry profiles and genetic interconnectedness. The Log Coffin-associated genomes can be modelled as an admixture between Hòabìnhian hunter-gatherer-, Yangtze River farmer-, and Yellow River farmer-related ancestry. This indicates different influence spheres from Bronze and Iron Age individuals from northeastern Thailand as reflected by cultural practices. Our analyses also identify close genetic relationships within the sites and more distant connections between sites in the same and different river valleys. In combination with high mitochondrial haplogroup diversity and genome-wide homogeneity, the Log Coffin-associated groups from north-western Thailand seem to have been a large, well-connected community, where genetic relatedness played a significant role in the mortuary ritual.

A vast number of caves intersperse the limestone karst formations of the southern Shan Hills in the northwestern part of Thailand. These diverse habitats, dominated by deciduous forests and controlled by tropical monsoon climate, provided a favourable setting for Palaeolithic hunter-gatherers and subsequent human settlements[1–3]. While the first lithic tools in northern Thailand were dated to 550 thousand years (ka) ago[4], excavations of hunter-gatherer sites suggest modern humans occupied the region from at least 35 ka ago[5], where high environmental heterogeneity allowed this subsistence strategy and so-called 'Hòabìnhian' stone artefact technology to persist for over 20 ka[6]. As cultural and genetic variation within the hunter-gatherers is only beginning to be uncovered[1,7,8], archaeological and archaeogenetic

[1]Department of Archaeogenetics, Max Planck Institute for Evolutionary Anthropology, Leipzig, Germany. [2]Department of Biology, Faculty of Science, Naresuan University, Phitsanulok, Thailand. [3]School of Computer and Mathematical Sciences, University of Adelaide, Adelaide, SA, Australia. [4]Institute for Archaeological Sciences, Archaeo- and Palaeogenetics, University of Tübingen, Tübingen, Germany. [5]Senckenberg Centre for Human Evolution and Palaeoenvironment, University of Tübingen, Tübingen, Germany. [6]Department of Evolutionary Genetics, Max Planck Institute for Evolutionary Anthropology, Leipzig, Germany. [7]Université Lyon 1, CNRS, Laboratoire de Biométrie et Biologie Evolutive, UMR 5558 Villeurbanne, France. [8]Department of Archaeology, Silpakorn University, Bangkok, Thailand. [9]The Prehistoric Population and Cultural Dynamics in Highland Pang Mapha Project, Princess Maha Chakri Sirindhorn Anthropology Centre, Bangkok, Thailand. ✉e-mail: selina_carlhoff@eva.mpg.de; shoocongdej_r@su.ac.th; krause@eva.mpg.de

studies have chronicled large-scale genetic and cultural shifts in local hunter-gatherer groups with the expansion of early sedentary food producers from East Asia[9–11]. Many habitational sites spreading along south-flowing river valleys appear in the archaeological record around 4.5–4 ka ago. They share inhumation rituals, pottery styles, and domestic animals such as dogs, pigs, and cattle[11–13]. Genetic evidence indicates that their inhabitants harboured both Neolithic southern East Asian- and local hunter-gatherer-related ancestry[7,8,10,14].

Based on the current evidence found across Thailand, it appears that the expansion of Neolithic material culture and technologies involved different migration routes and groups of people. One route may have followed the Salween river in the northwest, another along the Red and Mekong rivers on the northeastern and eastern coast of Thailand[14]. However, little is known about the genetic diversity in the northwestern corridor. Nowadays, highland Pang Mapha, a region in the Mae Hong Son province in northwestern Thailand, located near the western Salween river, is inhabited by many ethnic groups, including the Shan, Karen, Red and Black Lahu, Chinese, Lawa, Lisu, Hmong, and Thai[2], speaking Sino-Tibetan, Tai-Kadai, Hmong-Mien, and Austroasiatic languages[15]. Early speakers of these language families might have interacted, or even been part of the same ancient communities in southern China 3–1 ka ago[16].

Previous studies have suggested that the different ethnic groups, called 'Baiyue tribes' by the Han, might be connected to the Hanging Coffin and Log Coffin archaeological cultures of southern China and northwestern Thailand, respectively[17]. In highland Pang Mapha, over 40 caves and rock shelters with large wooden coffins, dating between 2.3 and 1 ka ago, have been discovered in five river valleys. The Log Coffin culture reflects one of the regional variations in mortuary practice during the Late prehistory of mainland Southeast Asia[18]. Coffins were cut from a single tree and feature distinct carvings at the head and foot ends, which may reflect societal beliefs, the status of the

deceased, the skill of the coffin's maker, or indicate family or clan cemeteries[19–22]. Other artefacts recovered from these caves include pottery sherds and faunal remains, and iron tools and bronze objects place them in the Iron Age of the region[2,3]. Dendrochronological analysis and radiocarbon dating suggest use as cemeteries for several hundred years[23,24]. The conspicuousness of the wooden coffins has generated a lot of scientific and public interest but due to many instances of looting and the lack of habitation sites associated with the cemeteries, detailed insights into the life of the associated people and how they are connected to previous and present-day inhabitants of the region remain challenging. Since DNA preservation is poor in Southeast Asia, to date, only a few mostly low-coverage genomes associated with log coffins excavated at Long Long Rak[8] are available (Fig. 1). However, higher coverage data from a larger number of sampled individuals would be desirable, as this would enable more fine-scale analyses of genetic relatedness, such as shared identity-by-descent (IBD) blocks[25], community connectivity, and cemetery structure.

Here, we present the genetic and archaeological analyses of 33 newly-sequenced individuals from five Iron Age Log Coffin culture sites in northwestern Thailand. We evaluate genetic ancestry profiles of ancient individuals from Thailand and the surrounding regions by focussing on the differences in hunter-gatherer- and Neolithic farmer-related ancestries. Additionally, we compare the ancient genetic signatures with those of the present-day inhabitants of the region and investigate genetic relatedness and connectivity within and between different burial sites.

## Results

For this study, we assessed 64 pieces of skeletal material from eight archaeological sites in northwestern and northern Thailand, including 42 petrous parts of the temporal bone and 22 teeth, of which 60 were sampled for ancient DNA (Supplementary Fig. 1, Supplementary Data 1;

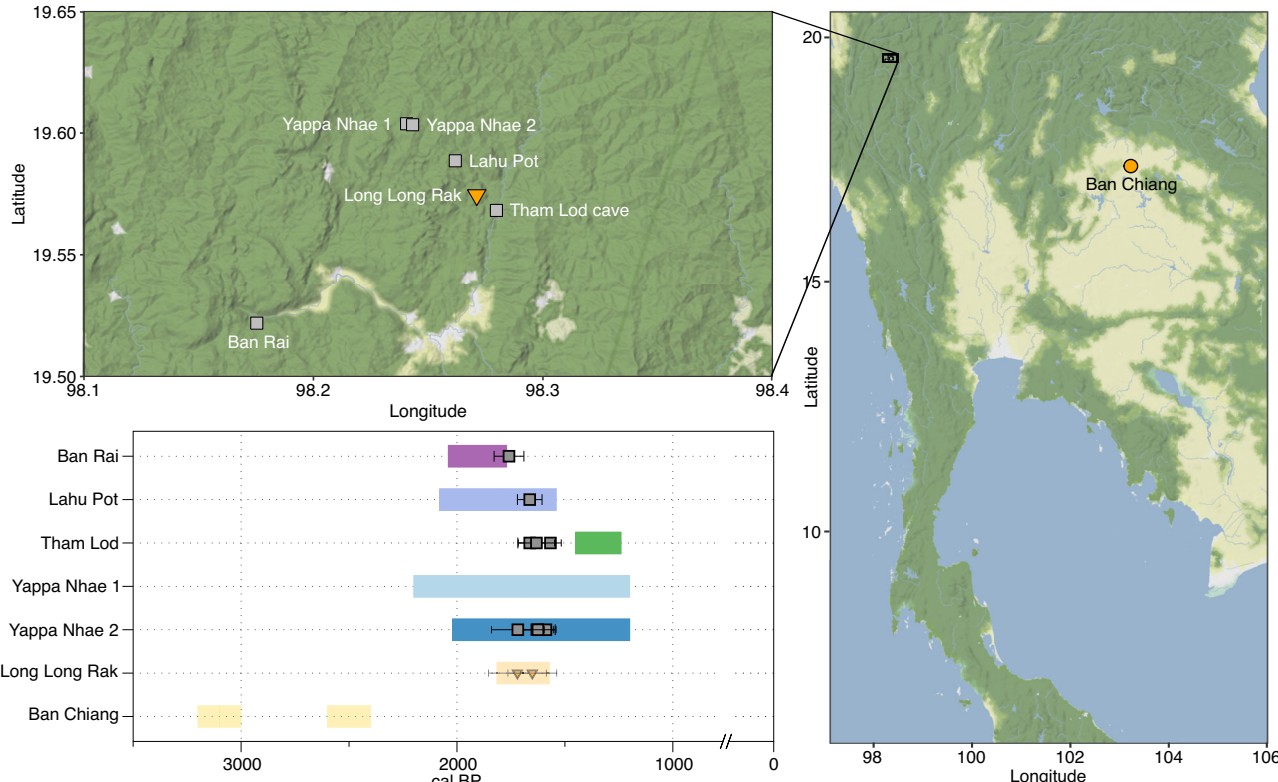

**Fig. 1 | The location and age of studied ancient individuals from Thailand.** Previously published sites are marked in orange[8,10], bars denote archaeological age and symbols denote direct calibrated $^{14}$C dates, plotted with ggmap v.4.0.0[69] and

DataGraph v.4.4. The error bars show one standard error. Map tiles by Stamen Design (CC BY 3.0), data by OpenStreetMap (ODbL). Source data are provided as a Source Data file.

see Supplementary Notes for descriptions of all sites). We were able to recover genetic material suitable for population genetic analyses from the sites Ban Rai ($n = 1$), Lahu Pot ($n = 1$), Tham Lod ($n = 3$), Yappa Nhae 1 ($n = 2$), and Yappa Nhae 2 ($n = 26$). A subsample of petrous bones was directly [14]C dated to 1.6–1.8 ka calBP (Fig. 1, Supplementary Data 2).

After double-stranded library preparation and shotgun screening, we evaluated the authenticity and preservation level of ancient DNA fragments by considering short DNA fragment length, the percentage of endogenous human DNA, and the elevation of C→T mis-incorporations at the 5' end of DNA strands[26] (Supplementary Discussion, Supplementary Fig. 2, Supplementary Data 1). Based on these criteria, we selected 33 DNA libraries to be enriched for ~1.2 million (1240K capture) single nucleotide polymorphisms (SNPs) using in-solution DNA hybridization capture[27,28]. Final coverage was variable, with a median number of 489,787 SNPs on the 1240K panel (range: 52,907−935,000 SNPs) and minimal contamination levels (Supplementary Data 1). Seventeen individuals with high X-rates most likely possessed the karyotype XX and YPN020 likely XXX, while 15 individuals had elevated Y-rates indicating XY (Supplementary Data 1). The main mitochondrial haplogroups were F1a1a, F1f, M7b1a1, and N8*, while the Y-haplogroups were mostly O1b and O2a, in addition to N1b and C2b (Supplementary Fig. 3, Supplementary Data 1). A detailed analysis of uniparental markers will be provided in a forthcoming study.

We assessed genetic relatedness with KIN v.3.1.3[29] and determined that two genomes were likely from the same individual (YPN032-YPN033) within Yappa Nhae 1, while there was one parent-child pair (YPN014-YPN021) within Yappa Nhae 2 (Supplementary Data 3). We also determined four second-degree relationships, including one grandparent-grandchild (YPN010-YPN011) and three avuncular or half-sibling relationships within and between Yappa Nhae 1 and 2. Additionally, we analyzed the published genomes from Long Long Rak and identified two as the same individual (Th519-Th703) and a potential sibling pair (Th530-Th531). Based on runs of homozygosity[30] (Supplementary Discussion), we also assigned four individuals from Yappa Nhae 2 as offspring of closely related parents (YPN001, YPN002, YPN006, YPN022; Supplementary Fig. 4).

To study more distant genetic relationships, we detected shared IBD blocks >20 cM after imputation between high-coverage individuals from Lahu Pot, Tham Lod, and Yappa Nhae 2 (Supplementary Discussion). We found the highest amount of IBD sharing within Yappa Nhae 2, with the highest total IBD block length shared between YPN010 and YPN011 (Supplementary Fig. 5). A cluster of individuals from Yappa Nhae 2 (YPN001, YPN006, YPN012, YPN027, YPN030) were all closely related to each other, while other individuals from the site were only distantly related to one or two individuals of this group. YPN003 and YP007, who shared 825 cM of IBD blocks and MT- and Y-haplogroups, did not share long IBD blocks or haplogroups with any other

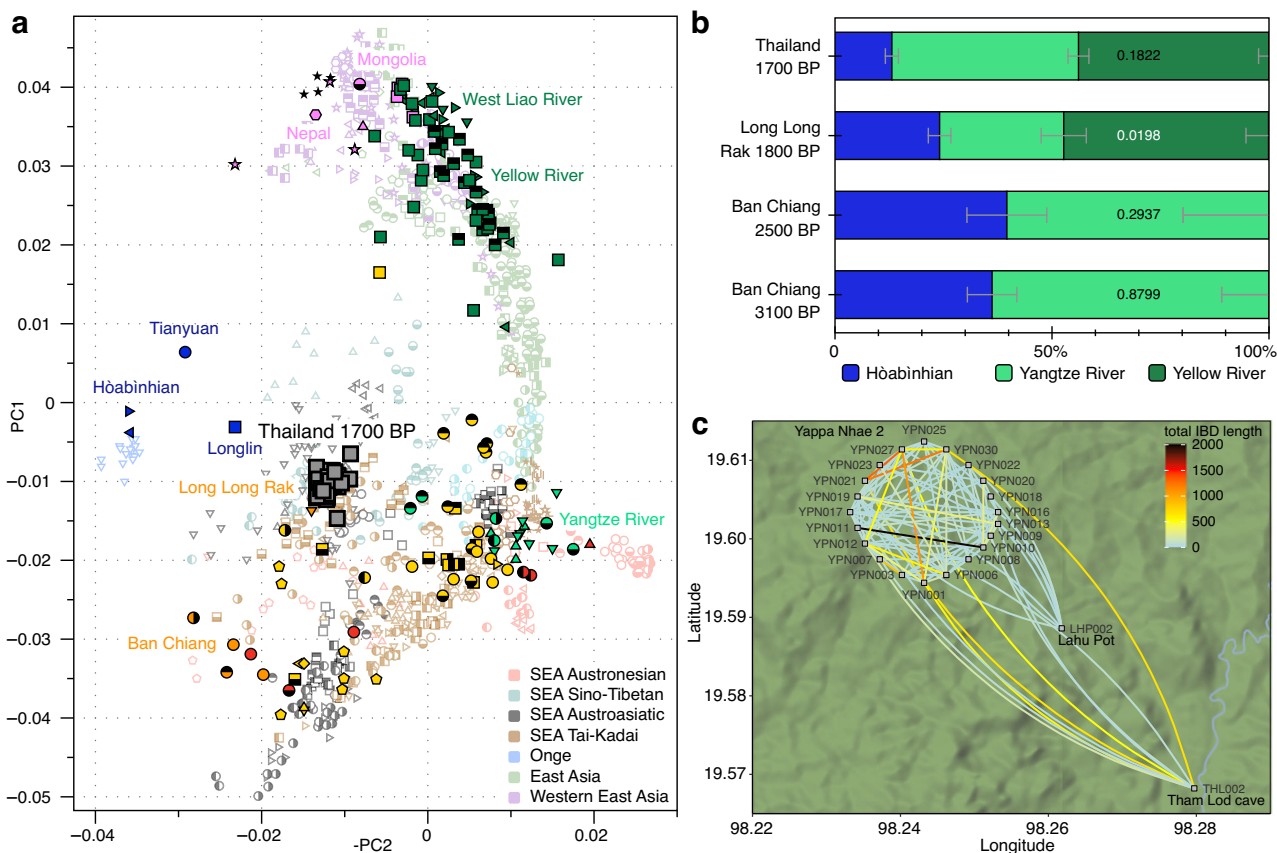

**Fig. 2 | Regional and local connectivity of the Log Coffin-associated genomes.**
**a** PCA calculated on present-day individuals from East and Southeast Asia[8,31–35], individuals from Southeast Asia (SEA) are coloured by language family, ancient individuals are projected[7–10,32,36–38], full legend is available in Supplementary Fig. 5. **b** Admixture proportions modelling Log Coffin culture-associated individuals from northwestern Thailand, including Long Long Rak[8], and Bronze and Iron Age individuals from Ban Chiang, northeastern Thailand[10] as a combination of ancient Hòabìnhian[8], Yellow River-[38] and Yangtze River-associated[9] ancient individuals, plotted values denote the best-fitting proportions as calculated by qpAdm v.1520[35],

error bars in grey show ± 1 standard error as calculated with block jackknife in qpAdm v.1520[35], $p$-values are printed within the bars, $p$-values in white indicate an unsatisfactory fit ($p < 0.05$). **c** Shared IBD block lengths between newly generated ancient genomes plotted in the geographical location of the respective sites; individuals from Yappa Nhae 2 have been artificially spread out to show within-site connections (Supplementary Data 4), plotted with ggmap v.4.0.0 using map tiles by Stamen Design (CC BY 4.0) and data by OpenStreetMap (ODbL). Source data are provided as a Source Data file.

individuals. Some individuals from Yappa Nhae 2 also shared distant relationships with those from Tham Lod and Lahu Pot (Fig. 2c). Still, the frequency and length of shared blocks between pairs of sites differed, with a maximum of 12 blocks of 668.9 cM total shared by Tham Lod-Yappa Nhae 2 (THL002–YPN030) compared to a maximum of two blocks of 57.8 cM total shared by Lahu Pot-Yappa Nhae 2 (LHP002–YPN022) and no shared IBD blocks between Lahu Pot and Tham Lod (Supplementary Fig. 5, Supplementary Data 4).

We performed a principal component analysis to assess the genetic similarities between the newly-sequenced individuals and ancient and present-day individuals from the region[7–10,31–38] (Fig. 2a, Supplementary Fig. 6, Supplementary Discussion). The Log Coffin-associated individuals formed a cluster which also contained individuals from the previously studied Log Coffin site Long Long Rak[8] and fell on top of present-day individuals from Thailand[31] (Sino-Tibetan-speaking Padaung Karen and Austroasiatic-speaking Eastern Lawa, Western Lawa, and Mon). This position was distinct from other ancient individuals, including Bronze Age and Iron Age individuals from Ban Chiang in northeastern Thailand[10].

To formally investigate the genetic processes underlying the observed positions of the individuals in PCA space, we calculated a series of $f$-statistics (Supplementary Discussion). When testing the relationship between the newly sequenced individuals and Bronze Age individuals from northeastern Thailand compared to ancient genomes from East Asia ($f_4$(Mbuti, ancient East Asia; Bronze Age Ban Chiang, Log Coffin)), we observed a significant affinity of the Log Coffin-associated individuals to ancient groups from northern East Asia, particularly from the Yellow River valley[38], Mongolia[32], and Nepal[37] (Supplementary Fig. 7). Based on these results, we were able to model the newly-generated genomes, as well as the previously published ones from Long Long Rak[8], as a two-way admixture of Bronze Age individuals from Ban Chiang, and Late Neolithic/Iron Age Yellow River or West Liao River groups using qpAdm v.1520[35] ($p = 0.4789$, Supplementary Fig. 11, Supplementary Discussion). In contrast, these models did not fit for individuals from the Iron Age in Ban Chiang (nested $p = 0.4665$). When dividing the ancestry of Bronze Age individuals into hunter-gatherer- and early farmer-related components[7–10], the Hòabìnhian-associated individual from Laos was determined to be the best fitting hunter-gatherer-related source for the Log Coffin-associated individuals, while both Hòabìnhian- and Longlin-related individuals resulted in similarly well-fitting models for the individuals from Ban Chiang (Supplementary Fig. 12). We successfully modelled the newly-generated Log Coffin-associated genomes as a three-way admixture between $13.0 \pm 1.5\%$ Hòabìnhian-, $43.0 \pm 2.4\%$ southern East Asian- and $44.0 \pm 2.1\%$ Upper Yellow River-related ancestry ($p = 0.8799$, similar to Laos 2-10 ka and Late Neolithic and historic Vietnam (Figs. 2b, 3a, Supplementary Fig. 13, Supplementary Data 5). Conversely, no Upper Yellow River-related ancestry was required for modelling the Bronze Age and Iron Age individuals from Ban Chiang, as well as the Neolithic individuals from Man Bac in Vietnam (Figs. 2b, 3a, Supplementary Data 5). We also observed slight variation in the relative amounts of these ancestries between the Log Coffin-associated individuals from northwestern Thailand (Supplementary Fig. 14).

We then tested for genetic continuity from the ancient individuals to present-day groups from Thailand[31]. While some present-day groups clustered with the ancient individuals in PCA space (Fig. 2a, Supplementary Fig. 6), ADMIXTURE v.1.3.0[39] analysis showed that most present-day groups contained more and other combinations of ancestry components compared to the ancient genomes (Fig. 3b, Supplementary Figs. 15, 16, Supplementary Discussion). Further testing with $f_4$(Mbuti, present-day Thailand; Bronze Age Ban Chiang, Log Coffin) confirmed a generalized affinity of present-day groups to the newly reported ancient genomes (Supplementary Fig. 8), when compared to Bronze Age Ban Chiang, but with no attraction to any specific group or language family (Supplementary Figs. 9, 10).

## Discussion

Here, we present genome-wide data of 33 individuals associated with the Iron Age Log Coffin culture of northwestern Thailand. The low level of long runs of homozygosity and the high mitochondrial haplogroup diversity indicate that a large community lived in the highlands, which is supported by the high number of log coffin sites found in the area[18,21]. From a population genetics perspective, the Log Coffin-associated individuals appear to be genetically homogeneous with little variation in PCA position and admixture components (Fig. 2a, Supplementary Figs. 6, 14). This pattern indicates older genetic contact with genetically-distinct communities or a large amount of genetic exchange within the recently admixed community. The Log Coffin-associated communities in northwestern Thailand retained ancestry from Hòabìnhian-associated groups, whereas 8 ka-old groups in southern China carry Longlin-related ancestry and younger groups from Guangxi and Bronze Age Vietnam do not carry either hunter-gatherer-related component (Supplementary Data 5). This indicates multiple interactions of incoming groups with local hunter-gatherers, as well as movement with little local admixture, resulting in a diverse (post-) Neolithic genetic landscape in Southeast Asia. This is further corroborated by the occupational history of some Log Coffin-associated caves reaching back into the Late Pleistocene and Early Holocene[3,5] and underlines the importance of understanding exactly when and how these interactions and transitions took place. However, poor DNA preservation in the region makes DNA retrieval from older periods particularly challenging.

At the regional level, we identify at least two distinct ancestry profiles in ancient individuals from Thailand. The Log Coffin-associated individuals from the highlands show a large additional genetic component related to groups from the Yellow River valley, which is not present in Bronze Age (3.1 ka) and Iron Age (2.5 ka) individuals from Ban Chiang in northeastern Thailand[10]. The genetic results are indicative of different spheres of influence for these groups during the Iron Age which is paralleled by the similarity of wooden coffin burials in northwestern Thailand, the central highlands of Vietnam, and southern China[21], compared to the burials underneath residence structures in northeastern Thailand[40]. Additionally, palaeoproteomic studies indicate a strongly $C_3$-based diet for Log Coffin-associated individuals from Long Long Rak, likely consisting of rice and freshwater fish among other food items, while the diets of Neolithic and Iron Age northeastern Thailand groups were more diverse, but depleted in $C_3$ food sources[41]. Although an analysis of mtDNA data concluded that the Log Coffin culture arrived via cultural transmission from Yunnan Province, China[17], the newly generated autosomal data shows an external genetic influence before or with the introduction of log coffins to northwestern Thailand. This differentiation could also reflect different migration routes of different Neolithic groups, with highland Pang Mapha being part of the northwestern route along the Salween river and mountain ranges.

The substantial gene flow into the studied Log Coffin communities from groups carrying Yellow River-related ancestry has parallels in other parts of East and Southeast Asia (Fig. 3a). Detected in ancient individuals from Fujian Province, China, after the Late Neolithic[9], this component arrived in Guangxi Province, China, between 6.4 and 1.5 ka ago[7] and is later present in Neolithic and Bronze Age individuals (4.2–2.1 ka) from Malaysia, Myanmar, Laos, and Vietnam[7,8]. Recently published ancient human genomic data also showed that Upper Yellow River-related ancestry contributed >80% to some early and present-day Tibetan Plateau groups, indicating genetic links before the introduction of barley agriculture onto the Plateau[42]. Since present-day groups of the Chinese Central Plain form a genetic cline between Upper Yellow River- and southern East Asian-related ancestry, the former has been associated with the spread of Sino-Tibetan languages from their homeland along the Upper Yellow River during the late Cishan/early Yangshao period (7.4 ka) to mainland Southeast Asia[32,43].

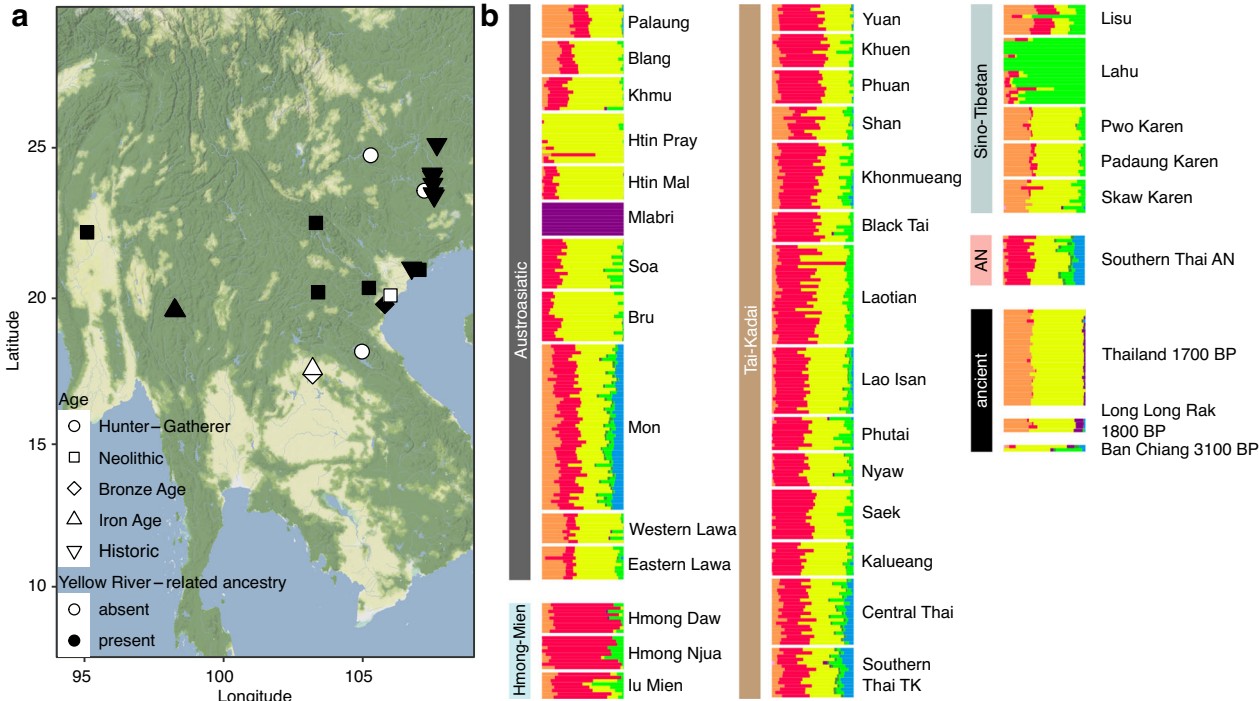

**Fig. 3 | Differential affinities of ancient and present-day genomes. a** Map of mainland Southeast Asia showing the presence (black) and absence (white) of northern East Asian ancestry in ancient individuals[7,8,10] (Supplementary Data 5), plotted with ggmap v.4.0.0[69], map tiles by Stamen Design (CC BY 3.0), data by OpenStreetMap (ODbL); **b** ADMIXTURE components of ancient and present-day groups from Thailand[8,10,31], represented at K = 8, separated by language family (AN = Austronesian, TK = Tai-Kadai; Supplementary Fig. 16). Source data are provided as a Source Data file.

Bronze Age and Iron Age Ban Chiang from northeastern Thailand and Neolithic Man Bac from northern Vietnam stand out in their lack of this genetic component, potentially pointing towards a temporal pattern, where this component only arrived in Thailand and Vietnam later, or perhaps cultural boundaries limited further propagation. Additional ancient genomes, in combination with novel admixture modelling and dating techniques, will help to date the introduction of this northern East Asian component, and understand the distribution and spread of this ancestry throughout East and Southeast Asia in the future.

Although a study of the mitochondrial haplogroup frequency spectra identified present-day Khon Mueang and Thai Yuan as most similar to ancient Log Coffin-associated individuals from northwestern Thailand[17], on a genome-wide level, present-day groups[31] show a generalized affinity to the ancient individuals from northwestern Thailand, but also harbour additional ancestry components (Fig. 3b, Supplementary Fig. 16). This indicates that significant changes in the gene pool took place after 1700 BP, including migrations from Laos, Myanmar, and southern China in historical times[44,45]. Still, groups in northwestern Thailand retain a level of genetic differentiation from other regions in Thailand[31,46,47].

Generating genomic data from many individuals associated with the same archaeological complex and recovered in close proximity also allowed detailed investigations into the social structure of an ancient Southeast Asian community. At the Log Coffin-associated sites, we identified several close genetic relationships between sampled individuals from Yappa Nhae 1 and 2, situated in the Mae Lana valley, and within the previously-published genomes from Long Long Rak. Furthermore, individuals from Lahu Pot and Tham Lod, located in the Lang river valley, and Yappa Nhae 2 shared long IBD blocks, suggesting close connections and exchange between the communities from different river valleys. Individuals from Yappa Nhae 2 and Tham Lod shared 3rd to 6th degrees of relatedness, while Yappa Nhae 2 and Lahu Pot were more distantly related through 5th to 6th degrees of relatedness. Present-day groups in the region practise both matri- and

patrilocality traceable through uniparental markers[45], but analysis of the uniparental markers suggests connections of the ancient related individuals through the maternal (e.g., YPN014-YPN021) and paternal lines (e.g., YPN014-YPN027-YPN030) within Yappa Nhae 2. Several 1st and 2nd degree relationships within Yappa Nhae and Long Long Rak, in addition to the six individuals from Yappa Nhae 2 and one from Tham Lod being more closely related to each other compared to other individuals from the site, could indicate a tendency for genetic relatives to be buried in the same cave. This corresponds with a social network analysis based on the coffin handles that suggested that the cemetery sites in each river valley were closely related and the respective communities interacted across the river valleys[48,49]. Since the two 1st degree-related individuals at Long Long Rak were also buried in the same coffin[8], the variety of the coffin handles may also reflect the diversity of families living in highland Pang Mapha from different periods. Detailed analysis of the burial placement in relation to genetic relatedness at the other studied sites is inhibited by later disturbance of the remains. This demonstrates the value of incorporating archaeogenetic results into archaeological findings and the in-depth study of single sites and archaeological complexes to reveal more about the lives and beliefs of past communities. Radiocarbon dating and genomic analyses of other sites with log coffin burials will help to further resolve the social structure of these communities, and to study the trans-regional connections and the spread of this mortuary practice.

## Methods

Permits for archaeological excavations, export, and archaeogenetic analyses of the analysed individuals were issued by the Fine Arts Department of Thailand under permit numbers 497/2561 and WTh-0417-236. After completion of laboratory analyses in Germany, all human remains were repatriated to the Fine Arts Department of Thailand. In terms of capacity building, our project has actively participated in undergraduate and doctoral research endeavours focused

on the field of archaeology at the Faculty of Archaeology, Silpakorn University, as well as genetic anthropology within the Faculty of Science at Khon Kaen University. In order to foster community engagement, the students and teachers from Tham Lod School have undertaken visits to the Long Long Rak cave both during and after the excavation activities. These visits have provided them with the opportunity to provide valuable feedback pertaining to the implementation of sustainable conservation practices. Finally, the project's outreach endeavors have effectively communicated research findings to three distinct audiences. Firstly, the Tham Lod community has been consistently informed about our research outcomes, including the DNA analysis. Secondly, the academic community has been reached through the dissemination of our findings in scientific journals and through online broadcasts of academic conferences. Lastly, the general public has been engaged through the publication of our research in newspapers, popular magazines, television programs, and social media platforms.

We assessed the preservation of 64 pieces of skeletal material from seven archaeological sites and selected 42 petrous bones and 22 teeth for ancient DNA analysis. Ancient DNA sampling, extraction, library preparation, and indexing were performed in a dedicated clean room for ancient DNA at the Max Planck Institute for the Science of Human History (now MPI of Geoanthropology) in Jena, Germany. We obtained bone powder from 40 petrous parts of the temporal bone by cutting along the margo superior partis petrosae (crista pyramidis) and drilling near the cochlea[50] and from 22 teeth by cutting along the enamel-dentin junction and drilling from the pulp (https://doi.org/10.17504/protocols.io.bqebmtan). The hardest part of the petrous bone was missing for two samples, and another two elements were too soft to sample successfully. Therefore, these samples were not further processed. A selection of petrous bones from sampled individuals was directly dated at the Klaus-Tschira-Archäometrie-Zentrum, Mannheim, Germany.

We extracted DNA following a modified version of the Dabney protocol[51] (https://doi.org/10.17504/protocols.io.baksicwe). The sample was digested in a mixture of EDTA, UV-irradiated High Performance Liquid Chromatography-grade $H_2O$ (UV-$H_2O$), and Proteinase K for approximately 24 hrs while rotating in an incubator at 37 °C. After centrifugation to remove the remaining bone material from solution, the supernatant was transferred into a binding buffer (GuHCl, UV-$H_2O$, Isopropanol) and then into a silica column (High Pure Viral Nucleic Acid Kit; Roche). The column was washed twice using the High Pure Viral Nucleic Acid Kit wash buffer (Roche) and eluted in a 100 µl TE-buffer containing 0.05% Tween. An extraction blank to check for cross- and background contamination and a positive control were added to this step and processed in parallel with the sample.

From the resulting extracts, 25 µl were used to build double-stranded libraries after partial Uracil-DNA Glycosylase treatment[52]. llumina adapters were ligated to the fragments using the Quick Ligation Kit (NBE) and the solution purified with a MinElute kit (QIAGEN). The DNA copy number of a 2 µl aliquot of the library was quantified by DyNAmo SYBR Green qPCR (Thermo Fisher Scientific) using IS7/IS8 primers on the LightCycler 480 (Roche)[53]. Based on those values, the library was double-indexed with unique index combinations[54] and amplified using PfuTurbo DNA Polymerase (Agilent; https://doi.org/10.17504/protocols.io.bvt8n6rw). Then it was purified over MinElute columns, eluted in 50 µl TE-buffer containing 0.05% Tween, and an aliquot quantified with IS5/IS6 primers using DyNAmo SYBR Green qPCR (Thermo Fisher Scientific) on the LightCycler 480 (Roche)[53]. The remaining solution was further amplified with Herculase II Fusion DNA Polymerase (Agilent; https://doi.org/10.17504/protocols.io.beqkjduw), purified again, quantified on the 4200 TapeStation System (Agilent), and diluted to 10 nM for shotgun sequencing. Throughout library preparation, a new blank, as well as the extraction blank, were taken along.

The prepared libraries were shotgun-sequenced for a depth of 2-10 million reads on an Illumina HiSeq 4000 using a 75 bp single-read configuration. The resulting reads were demultiplexed according to expected index combinations. Further processing was done in nf-core/eager v2.4.0[55] (https://nf-co.re/eager) using Nextflow v21.04.0[56] (see Supplementary Methods for parameters).

After initial quality assessment, 33 libraries were further amplified with the IS5/IS6 primers and hybridized in-solution to oligonucleotide probe sets (Agilent) to enrich for a targeted set of 1,237,207 single nucleotide polymorphisms (SNPs) across the human genome (1240K capture)[27,28]. Due to low preservation, BRB002, LHP002, THL003, and YPN036 went through two rounds of 1240K-capture before sequencing. After sequencing the 1240K capture products to 13-73 million reads on the Illumina HiSeq 4000 using a 75 bp single-read configuration, the reads were demultiplexed again. The resulting reads were aligned to the human reference genome (hg19) using mapping quality filter 30, duplicates removed, and genotyped with nf-core/eager v.2.4.0[55] (see Supplementary Methods for parameters).

We estimated nuclear contamination using ANGSD v.0.935[57] and karyotype with SexDetERRmine v.1.1.2[58] within nf-core/eager v.2.4.0[55]. We built mitochondrial consensus sequences using the export function in Geneious v.2019.2.3[59] setting "If no coverage call" and "Call {} if coverage <{}" to N/X with a coverage threshold of 5, "highest quality", and >50% Sanger heterozygotes. Mitochondrial haplogroups were determined with HaploGrep v.2.4.0[60] and mitochondrial contamination with ContamMix v.1.0-10[61]. For Y-chromosome assessment, we mapped the 1240K-enriched data to the human reference genome as described for the shotgun screening above. We then called all SNP positions on the Y-chromosome and compared the amount of ancestral and derived calls from the ISOGG database (https://isogg.org/tree/) to determine the Y-haplogroups of XY individuals[62].

We trimmed 2 bp off both ends of each read of the 1240K-captured sequences and genotyped these for the 1240K panel with pileupCaller v.1.4.0.5 (https://github.com/stschiff/sequenceTools) within nf-core/eager v.2.4.0[55]. We determined genetic relatedness with KIN v.3.1.3[29] and parental relatedness using runs of homozygosity in hapROH v.0.6[30] in Jupyter notebooks v.6.4.10. We ran the habsb_ind command with the provided down-sampled 1000 Genomes data as a reference panel and evaluated ROH on 22 chromosomes (chs=range(1,23)) with parameters matching our data (e_model="haploid", p_model="Eigenstrat", n_ref=2504, random_allele=True, readcounts=False, delete=False, logfile=True, combine=True).

For IBD calling, the ancient individuals were imputed to the 1000 Genomes positions[63] using ATLAS v.0.9[64] to call genotype likelihoods for the 33 newly-generated individuals. We called genotype likelihoods (method=MLE) for all positions from the 1000 Genomes Phase 3 release[63] after recalibrating the base quality scores according to post-mortem damage (length=50). These were then used for imputation with GLIMPSE v1.0.0[65], followed by extraction of the 1240K SNPs. We assessed imputation quality by counting the number of SNPs with genotype probabilities above 0.99 and excluded individuals with less than 600,000 well-imputed SNPs (BRB003, THL003, THL004, YPN002, YPN014, YPN018, YPN028, YPN031, YPN032, YPN033, YPN036). We then used ancIBD v.0.5[66] to calculate IBD blocks per individual from the 1240K-extracted data and compared these against each other, focusing on chunks with more than 220 SNPs and shared segments longer than 20 cM to identify relationships up to sixth-degree.

Initial population genetic analyses were conducted with smartpca v.16000[67] (lsqmode: YES, shrinkmode: YES) from EIGENSOFT v.7.2.1[67], where all ancient individuals were projected. We estimated proportions of ancestry using ADMIXTURE v.1.3.0[39] unsupervised on 4 parallel threads (-j 4), with random seed (-s ${RANDOM}) and cross-validation errors calculated (--cv). We estimated K = 2 through K = 15 and selected

the run with the highest likelihood out of five replicates per K. All $f_3$- and $f_4$-statistics were calculated with qp3pop v.651 and qpDstat v.980, respectively, while admixture proportions were estimated with qpAdm v.1520[35], all from AdmixTools v.7.0.2[35]. We used a rotating model selection approach[68], set inbreed: NO to be able to use single individuals in the model, and used Mbuti.DG as a SNP calculation basis in the references. The input files for qpAdm were generated with qpWrapper v.1.0.0 and results inspected with qpParser v.0.1.1 (https://github.com/TCLamnidis/qpWrapper).

### Reporting summary

Further information on research design is available in the Nature Portfolio Reporting Summary linked to this article.

## Data availability

The human remains are curated by R.S. at Department of Archaeology, Faculty of Archaeology, Silpakorn University, Bangkok, Thailand. The newly generated raw nuclear sequences are available at the European Nucleotide Archive under the accession number PRJEB59488. The human reference genome hg19 (GRCh37.p13) is available at the NCBI under accession number GCF_000001405.25, the newest version of the human reference genome hg38 (GRCh38.p14) is available at accession number GCF_000001405.40. Imputation reference data is available from the 1000 Genomes Project Phase 3. Previously published data used in principal component analysis and admixture modelling is available from the Poseidon Community Archive and at the Genome Sequencing Archive for Human under accession number HRA000451. Source Data for Fig. 2c and Supplementary Fig. 5B can be found in Supplementary Data 4. Source data for all other figures are provided with this paper.

## Code availability

The software used for all analyses is described in detail in the methods section and Supplementary Methods. R code for visualisation is available on GitHub.

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

## Acknowledgements

The research was funded by the Thailand Research Fund (RDG55H0006, W.K., R.S.) and RTA6080001 (W.K.), the ERC Starting Grant 'Waves' (ERC758967, S.C.), and the Max Planck Society (S.C., A.B.R., C.P., M.S., K.N., J.K.). W.K. was also funded by Institute of Suvanabhumi Studies, TASSHA (Thailand Academy of Social Sciences, Humanities and Arts), Ministry of Higher Education, Science, Research and Innovation, and Naresuan University under global and frontier research university fund (R2566C051). We thank Benchawan Phonprasoet and Siriluck Kanthasri for coordinating sample collection. We thank the laboratory and bioinformatics team at the Max Planck Institute for Evolutionary Anthropology, in particular Rita Radzeviciute and Antje Wissgott, for their excellent support, as well as Valentina Zaro for help during sampling.

## Author contributions

M.S., W.K., J.K., and R.S. conceived the study. R.S. led the archaeological excavations and surveys. W.K. organised sample collection. S.C. and K.N. sampled and extracted DNA and analysed the data with input from A.B.R., C.P., and M.S. S.C. and K.N. interpreted the data with critical input and contextualisation from R.S. and W.K. S.C. wrote the manuscript with critical input from K.N. and the remaining authors.

## Funding

## Competing interests

The authors declare no competing interests.
