## [Peer Review File · Nature Communications]

Genomic portrait and relatedness patterns of the Iron Age Log Coffin culture in northwestern ThailandReviewers' Comments:

Reviewer #1:

Remarks to the Author:

General Comments:

This is a significant research work worthy of publication in Nature Communications. It provides a novel set of genome-wide data encompassing 33 prehistoric human individuals from the well-defined 'log-coffin' cultural zone in the Pang-ma-pha district of Mae-hong-son province, northwestern Thailand. Spanning the period from approximately 2300 to 1000 years before present (BP), the culture is known for its unique mortuary practice of burying their dead in log coffins deposited in caves scattering over the Pang-ma-pha's karstic landscape. The people of the culture, perceived as late-prehistoric/iron age, are enigmatic in terms of their origin, demographic dynamics, and association with later inhabitants of the area, currently comprising diverse ethnic peoples affiliated with the Tai-Kadai, Austroasiatic, Sino-Tibetan, and Hmong-Mien linguistic families. The area was also evidently occupied prior to the log-coffin/iron age by earlier hunting-gathering people associated with the 'Hoabinhian' stone artifact technology, dating from at least around 12500 to 8000 years before present (BP).

The study made major discoveries through various genomic comparisons conducted on a broad scale encompassing East and Southeast Asia, and including analyses of both ancient and modern human genomes so far reported from these regions. Due to the lack of Hoabinhian-associated genomic data from Pang-ma-pha, the research did not provide an in-situ proof for continuity/discontinuity in the evolutionary terms from the earlier Hoabinhian to the later iron-age people. With employment of the Hoabinhian genomic data from elsewhere, however, the study was able to confirm presence of the Hoabinhian genetic components in the 'log coffin' human genomes possibly suggesting admixture between later Hoabinhian and ancestral 'log-coffin' peoples. The genetic diversity among the Pang-ma-pha's ancient genomes pertains the internal structure linking a number of them under specific degrees of kinship connection, both within and across the sites, suggesting the preference of sharing the same cemetery cave among blood-related families and the existence of kinship connections across the Pang-ma-pha's valleys during the period of concern. Integrated into the scaffolding portrait of genetic relationship established among the complete genomes derived from the modern samples, the diversity of the lower-coverage ancient genomes from Pang-ma-pha overlaps considerably with the diversities representing current inhabitants of the area, particularly those affiliated in the linguistic terms with Austroasiatic and Sino-tibetan.

Another comparison among the ancient samples possibly reveals genetic contribution to the 'log-coffin' genomes from the far-off earlier Yellow River farmers practicing millet cultivation, the contribution not noticeably detected in ancient genomes related with the neolithic farmers from Ban-chiang in northeastern Thailand and Man Bac in coastal northern Vietnam. All these southeast Asian ancient groups demonstrated the genomic profiles with significant genetic contributions from the neolithic farmers in southern East Asia, supposed to be the original homeland for rice cultivation. The study called for attention of the more elaborate but clearer picture of the genetic landscape related to the history and dynamism of human occupations in Southeast Asia, particularly on the issues related with expansions of the neolithic peoples from East Asia along with the spread of their corresponding farming cultures.

In summary, the study represents a significant advancement in archaeogenetic research related to Southeast Asian prehistory. It addresses the challenge of structuring the genomic relationships between prehistoric populations in East and Southeast Asia. Based on this groundwork, a sound hypothesis was proposed for a southwestward diffusion of the Yellow River farmers' gene pool from northern East Asia to be finally constituted in the genomes of the 'log-coffin' people in northwestern Thailand. This genetic diffusion was separate from those bringing the genetic components from the neolithic farmers in southern East Asia, probably along with the cultural spread associated with rice farming, to be widely harbored in early farmers of Southeast Asia.

Technical Comments:

(1) The authors' use of some ethnically-related terms could be confusing and warrants clarification. 'Thai' generally refers to the major ethnic group of the country of Thailand, itself also harboring other diverse ethnic groups. The term could not be directly used as an adjective for 'Thailand.' In this respect, 'ancient Thai individuals' (whose ethnic affiliation is unknown) and 'present-day Thai individuals (Karen Padaung, Lawa Eastern,)' could be rephrased respectively as 'ancient individuals from Thailand' and 'individuals from present-day Thailand (.....)'. Additionally, the ethnic names 'Karen Padaung' and 'Lawa Eastern/Western' should be rephrased as 'Padaung Karen' and 'Eastern/Western Lawa,' respectively. The authors should also ensure that there are no other similar circumstances that fall in the same category of problem.

(2) The authors have not provided a clear explanation of their criteria for designing the color scheme used in the PCA graphics. Some of the colors are difficult to distinguish from their contrasts, and it is unclear whether the colors systematically and consistently represent different geographic regions, cultures, linguistic affiliations, or any combinations of these. Moreover, the graphic resolution in both the manuscript and its supplement is inadequate. For example, the set of dark-colored symbols representing various ancient hunting and gathering groups in East and Southeast Asia lacks sufficient resolution to clearly distinguish between them.

(3) The evaluations related to the laboratory works on preparation and DNA extraction of the ancient samples, as well as the technical appropriateness of statistics and software tools utilized in advanced genomic comparisons, are not in the area of expertise of the reviewer."

Reviewer #2:

Remarks to the Author:

* What are the noteworthy results?

This is a very interesting and substantial study of genome-wide data from 33 individuals, which assesses ancestry and relatedness from an important log coffin sites in North-western Thailand. Interestingly, the authors point out that there are differences in ancestry between the individuals interred at this sites compared to other genomic information from individuals in Central and Northeast Thailand. This study is important in the social anthropological sense as it starts to look at genetic relatedness and interpretations of social structure and mortuary rites.

* Will the work be of significance to the field and related fields? How does it compare to the established literature? If the work is not original, please provide relevant references.

This work is significant for the field of archaeology in Southeast Asia and beyond. It is original and has not been undertaken in the region before.

* Does the work support the conclusions and claims, or is additional evidence needed?

Yes it does seem to, although some of the discussion could be teased apart with information from the bioarchaeology from the sites.

The aims are

* Are there any flaws in the data analysis, interpretation and conclusions? Do these prohibit publication or require revision?

No

* Is the methodology sound? Does the work meet the expected standards in your field?

I am not an expert in aDNA analysis but the methods and controls seem sound and rigorous.

* Is there enough detail provided in the methods for the work to be reproduced?

Yes

Some more minor points. It is stated in the abstract that the site is represented "by a unique mortuary practice known as Log Coffin culture" Technically this log coffin burial ritual is not unique but it found in neighbouring Cambodia and other countries. This needs to be edited slightly.

It is stated: "The low level of long runs of homozygosity and the high 179 mitochondrial haplogroup diversity indicate that a large community lived in the highlands, which is supported by the high number of log coffin sites found in the area" How big is big? What does the bioarchaeological work tell us about the minimum number of individuals in the sites and the age structure of those individuals to tell us something about fertility/population growth?

It is noted that it is thought that some of the people buried together within the sites are related to each other. Is there enough information to look at degree of relatedness in any more detail? My understanding is that there is the potential that some of the skeletal remains are 'loose' in that they are not subsurface at at least one of the sites. Therefore, how can you tell if they have been moved around or not post burial, or were the coffins closed/sealed prior to analysis? Unfortunately this part of the analysis seems a little haphazard, which I find with some aDNA studies when they start out without clear research questions and sampling strategy related to that research question. This is somewhat understandable given the sometimes poor preservation of aDNA in SE Asia and not knowing what data you will get.

Reviewer #3:

Remarks to the Author:

The manuscript "Genomic portrait and relatedness patterns of the Iron Age Log Coffin culture in northwestern Thailand" displays genome-wide data for 33 new individuals sampled in 5 sites associated with the Log Coffin culture in Mainland Southeast Asia. The manuscript describes high quality data and a battery of analyses that discerns various aspects of the genetic history of the Iron Age Log Coffin culture and MSEA prehistory in general. It also brings many unanswered questions (and currently unanswerable), that might be addressed in the future. I do not have any major concern with the manuscript.

A few points:

- 1) How does "high mitochondrial haplogroup diversity" and "genome-wide homogeneity" indicate that these were a large and well-connected community? This might sound to the casual reader that mtDNA displays high diversity and genome-wide low and both systems display opposite views.
- 2) Why do you consider that mtDNA diversity in the Log Coffin culture people does not show the trace of Yellow River ancestry that is so large in the genome-wide? It might be complex models as you mentioned. The cited mtDNA analysis might not be very conclusive. But given this opposite view between GW and mtDNA show you not consider a sex-biased ancestry?
- 3) On a similar note, you do not take any conclusions on the Y-chromosome lineages. Could you provide any discussion on those, or, if they are not informative at all, you should mention that.

The ò in Hòabinhian never shows up correctly throughout the manuscript.

Point-by-point response

Reviewer #1 (Remarks to the Author):

General Comments:

This is a significant research work worthy of publication in Nature Communications. It provides a novel set of genome-wide data encompassing 33 prehistoric human individuals from the well-defined 'log-coffin' cultural zone in the Pang-ma-pha district of Mae-hong-son province, northwestern Thailand. Spanning the period from approximately 2300 to 1000 years before present (BP), the culture is known for its unique mortuary practice of burying their dead in log coffins deposited in caves scattering over the Pang-ma-pha's karstic landscape. The people of the culture, perceived as late-prehistoric/iron age, are enigmatic in terms of their origin, demographic dynamics, and association with later inhabitants of the area, currently comprising diverse ethnic peoples affiliated with the Tai-Kadai, Austroasiatic, Sino-Tibetan, and Hmong-Mien linguistic families. The area was also evidently occupied prior to the log-coffin/iron age by earlier hunting-gathering people associated with the 'Hoabinhian' stone artifact technology, dating from at least around 12500 to 8000 years before present (BP).

The study made major discoveries through various genomic comparisons conducted on a broad scale encompassing East and Southeast Asia, and including analyses of both ancient and modern human genomes so far reported from these regions. Due to the lack of Hoabinhian-associated genomic data from Pang-ma-pha, the research did not provide an in-situ proof for continuity/discontinuity in the evolutionary terms from the earlier Hoabinhian to the later iron-age people. With employment of the Hoabinhian genomic data from elsewhere, however, the study was able to confirm presence of the Hoabinhian genetic components in the 'log coffin' human genomes possibly suggesting admixture between later Hoabinhian and ancestral 'log-coffin' peoples. The genetic diversity among the Pang-ma-pha's ancient genomes pertains the internal structure linking a number of them under specific degrees of kinship connection, both within and across the sites, suggesting the preference of sharing the same cemetery cave among blood-related families and the existence of kinship connections across the Pang-ma-pha's valleys during the period of concern. Integrated into the scaffolding portrait of genetic relationship established among the complete genomes derived from the modern samples, the diversity of the lower-coverage ancient genomes from Pang-ma-pha overlaps considerably with the diversities representing current inhabitants of the area, particularly those affiliated in the linguistic terms with Austroasiatic and Sino-tibetan.

Another comparison among the ancient samples possibly reveals genetic contribution to the 'log-coffin' genomes from the far-off earlier Yellow River farmers practicing millet cultivation, the contribution not noticeably detected in ancient genomes related with the neolithic farmers from Ban-chiang in northeastern Thailand and Man Bac in coastal northern Vietnam. All these southeast Asian ancient groups demonstrated the genomic profiles with significant genetic contributions from the neolithic farmers in southern East Asia, supposed to be the original homeland for rice cultivation. The study called for attention of the more elaborate but clearer picture of the genetic landscape related to the history and dynamism of human occupations in Southeast Asia, particularly on the issues related with expansions of the neolithic peoples from East Asia along with the spread of their corresponding farming cultures.

In summary, the study represents a significant advancement in archaeogenetic research related

to Southeast Asian prehistory. It addresses the challenge of structuring the genomic relationships between prehistoric populations in East and Southeast Asia. Based on this groundwork, a sound hypothesis was proposed for a southwestward diffusion of the Yellow River farmers' gene pool from northern East Asia to be finally constituted in the genomes of the 'log-coffin' people in northwestern Thailand. This genetic diffusion was separate from those bringing the genetic components from the neolithic farmers in southern East Asia, probably along with the cultural spread associated with rice farming, to be widely harbored in early farmers of Southeast Asia.

We thank the reviewer for their summary and assessment of our results.

Technical Comments:

(1) The authors' use of some ethnically-related terms could be confusing and warrants clarification. 'Thai' generally refers to the major ethnic group of the country of Thailand, itself also harboring other diverse ethnic groups. The term could not be directly used as an adjective for 'Thailand.' In this respect, 'ancient Thai individuals' (whose ethnic affiliation is unknown) and 'present-day Thai individuals (Karen Padaung, Lawa Eastern,)' could be rephrased respectively as 'ancient individuals from Thailand' and 'individuals from present-day Thailand (.....)'. Additionally, the ethnic names 'Karen Padaung' and 'Lawa Eastern/Western' should be rephrased as 'Padaung Karen' and 'Eastern/Western Lawa,' respectively. The authors should also ensure that there are no other similar circumstances that fall in the same category of problem.

We have made the appropriate changes across the manuscript, supplementary materials and respective figures (Padaung Karen, Pwo Karen, Skaw Karen, Eastern Lawa, Western Lawa).

(2) The authors have not provided a clear explanation of their criteria for designing the color scheme used in the PCA graphics. Some of the colors are difficult to distinguish from their contrasts, and it is unclear whether the colors systematically and consistently represent different geographic regions, cultures, linguistic affiliations, or any combinations of these. Moreover, the graphic resolution in both the manuscript and its supplement is inadequate. For example, the set of dark-colored symbols representing various ancient hunting and gathering groups in East and Southeast Asia lacks sufficient resolution to clearly distinguish between them.

We have adjusted the colour schemes of the PCAs presented in the manuscript and supplementary materials to better identify present-day and ancient groups relevant for the understanding of our results. Present-day individuals from Southeast Asia are now coloured by language family and all groups designated by a different combination of colour and shape. We have also reduced the number of ancient genomes in Figure 2 to only focus on those mentioned in the manuscript and re-plotted Figure 1 with more contrast and Figure 3a with more consistent shapes and colours.

(3) The evaluations related to the laboratory works on preparation and DNA extraction of the ancient samples, as well as the technical appropriateness of statistics and software tools utilized in advanced genomic comparisons, are not in the area of expertise of the reviewer."

The laboratory and statistical methods, such as principal component analysis, *f*-statistics, qpAdm and ADMIXTURE, follow the current standards in the field of

archaeogenetics and have been similarly employed in recently published papers (e.g. Penske et al. (2023) Early contact between late farming and pastoralist societies in southeastern Europe, *Nature*; Simões et al. (2023) Northwest African Neolithic initiated by migrants from Iberia and Levant, *Nature* 618, 550–556). Genetic blocks identical-by-descent have also been used to understand genetic relatedness in a recent publication (Rivollat et al. (2023) Extensive pedigrees reveal the social organization of a Neolithic community, *Nature*).

We thank the reviewer for their time and comments.

Reviewer #2 (Remarks to the Author):

* What are the noteworthy results?

This is a very interesting and substantial study of genome-wide data from 33 individuals, which assesses ancestry and relatedness from an important log coffin sites in North-western Thailand. Interestingly, the authors point out that there are differences in ancestry between the individuals interred at this sites compared to other genomic information from individuals in Central and Northeast Thailand. This study is important in the social anthropological sense as it starts to look at genetic relatedness and interpretations of social structure and mortuary rites.

* Will the work be of significance to the field and related fields? How does it compare to the established literature? If the work is not original, please provide relevant references.

This work is significant for the field of archaeology in Southeast Asia and beyond. It is original and has not been undertaken in the region before.

* Does the work support the conclusions and claims, or is additional evidence needed?

Yes it does seem to, although some of the discussion could be teased apart with information from the bioarchaeology from the sites.

The aims are

* Are there any flaws in the data analysis, interpretation and conclusions? Do these prohibit publication or require revision?

No

* Is the methodology sound? Does the work meet the expected standards in your field?

I am not an expert in aDNA analysis but the methods and controls seem sound and rigorous.

* Is there enough detail provided in the methods for the work to be reproduced?

Yes

We thank the reviewer for their evaluation of our results.

Some more minor points. It is stated in the abstract that the site is represented “by a unique mortuary practice known as Log Coffin culture” Technically this log coffin burial ritual is not

unique but it found in neighbouring Cambodia and other countries. This needs to be edited slightly.

We have removed the term “unique” from the abstract and rephrased a sentence in the introduction (lines 72-73).

It is stated: “The low level of long runs of homozygosity and the high 179 mitochondrial haplogroup diversity indicate that a large community lived in the highlands, which is supported by the high number of log coffin sites found in the area” How big is big? What does the bioarchaeological work tell us about the minimum number of individuals in the sites and the age structure of those individuals to tell us something about fertility/population growth?

Our initial surveys identified a minimum number of nine individuals at Yappa Nhae 1 of which five were likely adults, two juveniles and one baby. At Yappa Nhae 2, the minimum number of individuals accounted for 23, mentioned in the site descriptions within the supplementary materials. These consisted of 19 adults and four children. We have added the detailed distribution of osteological age groups to the supplementary materials.

The archaeological analysis of Yappa Nhae 2 suggested that the cemetery possibly represented a family, rather than the entire population. Since we were not able to identify the specific age of every individual, an analysis of age structure is not possible for this site. Similarly, a comparison of all Log Coffin sites to assess population growth and fertility lay outside the scope of this paper.

While the analysis of runs of homozygosity can allow estimates of the effective population size, i.e. the number of individuals reproducing at a certain time, the extrapolation to census size is challenging, as this is impacted by factors we cannot know, such as generation times and marital patterns. Therefore, we only report the effective population size in the Supplementary Materials.

It is noted that it is thought that some of the people buried together within the sites are related to each other. Is there enough information to look at degree of relatedness in any more detail? My understanding is that there is the potential that some of the skeletal remains are ‘loose’ in that they are not subsurface at at least one of the sites. Therefore, how can you tell if they have been moved around or not post burial, or were the coffins closed/sealed prior to analysis? Unfortunately this part of the analysis seems a little haphazard, which I find with some aDNA studies when they start out without clear research questions and sampling strategy related to that research question. This is somewhat understandable given the sometimes poor preservation of aDNA in SE Asia and not knowing what data you will get.

A more detailed analysis of burial placement in relation to genetic relatedness is indeed beset by disturbance of the archaeological sites by animals and humans. The skeletal remains were mostly loose and spread on the surface but clustered in the same area near the coffins. The coffins at most sites, including Yappa Nhae, were not sealed, with the exception of some coffins at Long Long Rak cave. As the least disturbed site, Long Long Rak has been studied in a lot of detail, including the social structure from an archaeological perspective (Samrit, Chonchanok (2020) *Social Network of Log Coffin Culture in Highland Pang Mapha, Mae Hong Son*, in *Archaeology of Pre Thai in Highland Pang Mapha, Mae Hong Son Province*, 95–112. Bangkok: Ruen Khew Printing). However, based on the comparative data from a regional survey of Log Coffin sites, it can be generally assumed that Yappa Nhae was regularly used by same cultural

groups. This is why we employed the help of aDNA analysis to answer whether these people were genetically related groups or not. We have added lines 293-295 to the manuscript explaining these circumstances.

On the genetics side, the specification of more distant relationships identified through analysis of genetic blocks identical by descent (IBD blocks), such as differentiation between great-grandparents, second cousins, etc., is more limited, due to the little predictable nature of redistribution and splitting of IBD blocks during the many recombination events separating second-, third-degree and more distantly related individuals. Due to the low number of genomes recovered from surrounding sites, distant within-site relatedness could only be assessed at Yappa Nhae 2. We thank the reviewer for their remarks and hope these comments provide some insights.

Reviewer #3 (Remarks to the Author):

The manuscript “Genomic portrait and relatedness patterns of the Iron Age Log Coffin culture in northwestern Thailand” displays genome-wide data for 33 new individuals sampled in 5 sites associated with the Log Coffin culture in Mainland Southeast Asia. The manuscript describes high quality data and a battery of analyses that discerns various aspects of the genetic history of the Iron Age Log Coffin culture and MSEA prehistory in general. It also brings many unanswered questions (and currently unanswerable), that might be addressed in the future. I do not have any major concern with the manuscript.

We thank the reviewer for their appraisal of our manuscript.

A few points:

1) How does “high mitochondrial haplogroup diversity” and “genome-wide homogeneity” indicate that these were a large and well-connected community? This might sound to the causal reader that mtDNA displays high diversity and genome-wide low and both systems display opposite views.

We believe that the high mitochondrial diversity and genome-wide homogeneity, coupled with low runs of homozygosity and IBD connections, are the result of a community with high internal genetic exchange. Low internal genetic exchange would result in few mitochondrial haplogroups dominating each site, sub-structure at the genome-wide level, higher runs of homozygosity and few IBD connections between sites. We have added the raw IBD data for segments >20 cM as Supplementary Table 4 to enable better replication.

2) Why do you consider that mtDNA diversity in the Log Coffin culture people does not show the trace of Yellow River ancestry that is so large in the genome-wide? It might be complex models as you mentioned. The cited mtDNA analysis might not be very conclusive. But given this opposite view between GW and mtDNA show you not consider a sex-biased ancestry?

Direct assessment of sex-biased admixture through the study of admixture proportions on the X-chromosome were unfortunately not possible, due to the absence of X-chromosome data for northern East Asian individuals described in the Supplementary Materials. We are currently working on a separate manuscript analyzing the mitochondrial and Y-chromosome lineages of the Log Coffin-associated individuals in detail and hope to address the reviewer’s comments about sex-biased admixture and haplogroup composition compared to Yellow River groups there.

3) On a similar note, you do not take any conclusions on the Y-chromosome lineages. Could you provide any discussion on those, or, if they are not informative at all, you should mention that.

A detailed discussion of Y-chromosome results will also be included in the forthcoming manuscript. We have added lines 120-121 to explain these circumstances.

The ò in Hòabinhian never shows up correctly throughout the manuscript.

We have adjusted all figures to include the correct spelling. We thank the reviewer for their contributions to this manuscript.

Reviewers' Comments:

Reviewer #1:

Remarks to the Author:

The authors have addressed the reviewer's comments by implementing appropriate revisions throughout the manuscript.

Reviewer #2:

Remarks to the Author:

All the comments I had on the manuscript have been addressed well by the authors

Reviewer #4:

Remarks to the Author:

General comment

The authors provide genome-wide data of 33 individuals from the well-defined 'log-coffin' cultural zone. Employing established population genetic techniques adapted for ancient DNA analysis, they not only identify the ancestral origins of these individuals but also investigate potential familial relationships among them. The results are important and relevant for the region and the data is presented properly. I do not see problems beyond the already stated by the three reviewers.

Assessment on Reviewer #3 answers comments.

1) How do "high mitochondrial haplogroup diversity" and "genome-wide homogeneity" indicate that these were a large and well-connected community? This might sound to the causal reader that mtDNA displays high diversity and genome-wide low and both systems display opposite views.

I concur with the authors' response, as the existing evidence does not provide any indications that would suggest an alternative conclusion. A high level of mitochondrial diversity per se does not indicate anything regarding population mobility. The genomic data (IBD) unequivocally supports the notion of substantial population mobility while simultaneously underscoring population stability at a broader level.

Despite the author's announcing that the data about uniparental markers is going to be published soon, a graphic representation of diversity would be appreciated.

2) Why do you consider that mtDNA diversity in the Log Coffin culture people does not show the trace of Yellow River ancestry that is so large genome-wide? It might be complex models as you mentioned. The cited mtDNA analysis might not be very conclusive. But given this opposite view between GW and mtDNA do you not consider a sex-biased ancestry?

The authors state that they lack on reference to compare, if no X chromosome data is available, then there is no way to do such comparisons. I agree with the authors that in the absence of X-chromosome data, it is not possible to model these ancestries with qpAdm or any other statistical method that would be useful for this research question.

The authors have explicitly stated their intention not to present additional results in this manuscript, leaving us with the sole recourse of relying on Y and mtDNA haplogroups for further insights. Consequently, there are no other avenues for exploration within the scope of this study.

3) On a similar note, you do not take any conclusions on the Y-chromosome lineages. Could you provide any discussion on those, or, if they are not informative at all, you should mention that.

This is answered with the same narrative as point 2. I must admit that I am a bit surprised by slicing the data in this way but if the authors justify that the uniparental markers per se can bring differentiated narratives then I am fine with it.

Point-by-point response

Reviewer #1 (Remarks to the Author):

The authors have addressed the reviewer's comments by implementing appropriate revisions throughout the manuscript.

We thank the reviewer for their contributions to this manuscript.

Reviewer #2 (Remarks to the Author):

All the comments I had on the manuscript have been addressed well by the authors

We thank the reviewer for their help in improving this manuscript.

Reviewer #4 (Remarks to the Author):

General comment

The authors provide genome-wide data of 33 individuals from the well-defined 'log-coffin' cultural zone. Employing established population genetic techniques adapted for ancient DNA analysis, they not only identify the ancestral origins of these individuals but also investigate potential familial relationships among them. The results are important and relevant for the region and the data is presented properly. I do not see problems beyond the already stated by the three reviewers.

Assessment on Reviewer #3 answers comments.

1) How do "high mitochondrial haplogroup diversity" and "genome-wide homogeneity" indicate that these were a large and well-connected community? This might sound to the causal reader that mtDNA displays high diversity and genome-wide low and both systems display opposite views.

I concur with the authors' response, as the existing evidence does not provide any indications that would suggest an alternative conclusion. A high level of mitochondrial diversity per se does not indicate anything regarding population mobility. The genomic data (IBD) unequivocally supports the notion of substantial population mobility while simultaneously underscoring population stability at a broader level.

Despite the author's announcing that the data about uniparental markers is going to be published soon, a graphic representation of diversity would be appreciated.

2) Why do you consider that mtDNA diversity in the Log Coffin culture people does not show the trace of Yellow River ancestry that is so large genome-wide? It might be complex models as you mentioned. The cited mtDNA analysis might not be very conclusive. But given this opposite view between GW and mtDNA do you not consider a sex-biased ancestry?

The authors state that they lack on reference to compare, if no X chromosome data is available, then there is no way to do such comparisons. I agree with the authors that in the absence of X-chromosome data, it is not possible to model these ancestries with qpAdm or any other statistical method that would be useful for this research question.

The authors have explicitly stated their intention not to present additional results in this manuscript, leaving us with the sole recourse of relying on Y and mtDNA haplogroups for further insights. Consequently, there are no other avenues for exploration within the scope of this study.

3) On a similar note, you do not take any conclusions on the Y-chromosome lineages. Could you provide any discussion on those, or, if they are not informative at all, you should mention That.

This is answered with the same narrative as point 2. I must admit that I am a bit surprised by slicing the data in this way but If the authors justify that the uniparental markers per se can bring differentiated narratives then I am fine with it.

We appreciate the reviewer's assessment of the concerns of all previous reviewers. We believe that a paper focusing solely on the uniparental markers will provide the right setting for comparisons with the extensive present-day reference data available from Thailand and the region. Additionally, the authors are not aware of thorough uniparental study of ancient individuals from the region and believe there are more insights to be gained from co-analyzing previously published and the newly produced ancient genomes from that perspective. To provide some more insight into the haplogroup assignments, we have added a graphic representation of mitochondrial and Y-haplogroup diversity as Supplementary Figure 3.

We thank the reviewer for their time and comments.